# MRI Study of Paraspinal Muscles in Patients with Amyotrophic Lateral Sclerosis (ALS)

**DOI:** 10.3390/jcm9040934

**Published:** 2020-03-28

**Authors:** Luca Diamanti, Matteo Paoletti, Umberto Di Vita, Shaun Ivan Muzic, Cristina Cereda, Elena Ballante, Anna Pichiecchio

**Affiliations:** 1IRCCS Mondino Foundation, 27100 Pavia, Italy; matteo.paoletti@mondino.it (M.P.); cristina.cereda@mondino.it (C.C.); elena.ballante01@universitadipavia.it (E.B.); anna.pichiecchio@mondino.it (A.P.); 2Department of Brain and Behavioral Sciences, University of Pavia, 27100 Pavia, Italy; umberto.divita01@universitadipavia.it (U.D.V.); shaunivan.muzic@gmail.com (S.I.M.)

**Keywords:** amyotrophic lateral sclerosis, muscle magnetic resonance imaging, biomarkers, electromyography, muscle damage

## Abstract

Background: the study of paraspinal muscles is pivotal for the diagnosis and staging of Amyotrophic Lateral Sclerosis (ALS), and is usually performed by electromyography. Objective: to evaluate the role of paraspinal muscle MRI as a diagnostic biomarker in ALS. Methods: we evaluated T1-w images of newly diagnosed ALS patients (*n* = 14), age-matched healthy controls (*n* = 11), patients affected by inflammatory myopathy (*n* = 10), and lumbar radiculopathy (*n* = 19), and compared them semiquantitatively by using the Mercuri Scale. Results: a significant difference in the appearance of the psoas muscle was observed between ALS patients and patients with radiculopathy (*p* = 0.003); after stratifying ALS patients into spinal and bulbar onsets, we found a significant difference in the appearance of the longissimus dorsi muscle between the spinal onset ALS subgroup and bulbar onset ALS subgroup (*p* = 0.0245), while no difference was found for multifidus (*p* = 0.1441), iliocostal (*p* = 0.0655), and psoas muscles (*p* = 0.0813) between the cohort subgroups. Conclusions: paraspinal T1-w MRI could help to distinguish spinal ALS patients from healthy and pathological controls. Specifically, the study of longissimus dorsi could play the role of a diagnostic ALS biomarker.

## 1. Introduction

The paraspinal muscles have an essential role in the stabilization of the spine, the conservation of the correct posture, and the fluency of the trunk movements [1]. Specifically, the more medially located multifidus muscle is pivotal for the spinal balance; whereas the erector spinae muscles, consisting of the longissimus dorsi and iliocostal, are more laterally located and are involved in the remaining functions cited above [1]. The appearance of paraspinal muscles vary from person to person due to different variables (i.e., sex, body weight, physical activity) [1]. The evaluation of these muscles can be helpful in the diagnosis of pathological conditions, though their clinical assessment may be difficult to perform. A rough clinical evaluation may be based on the ability to rise from the supine position or to raise the trunk from the prone position, but these are complex movements involving several muscle groups, and their clinical assessment may be further influenced by the presence of concomitant respiratory dysfunction. Clinical assessment helps in the classification of specific patterns of involvement (i.e., Pisa syndrome, camptocormia), whereas electromyography can be useful in the evaluation of superficial muscles at the thoracic or lumbar level, as well as in estimating the severity of the disease [2,3]. However, an adequate analysis of single muscles and deep muscles is not possible through these means. In the past years, muscle magnetic resonance imaging (MRI) has had an increasingly larger role in the study of paraspinal muscles in several neuromuscular pathologies (i.e., inflammatory and hereditary myopathies) [4,5], adding relevant data regarding selective fatty replacement and inflammatory changes of these muscles in a clinical context. The evaluation of paraspinal muscles is essential during the diagnostic phase and follow up of patients affected by Amyotrophic Lateral Sclerosis (ALS), in order to demonstrate the involvement of the thoracic domain and define the severity of the disease [6]. Few studies have been conducted on the role of paraspinal muscle MRI in ALS [7,8], and these are heterogeneous in terms of study design, MRI sequences, and results. For instance, a study by Jenkins et al. demonstrated a statistically significant difference in T2-w images of paraspinal muscles, between healthy controls (*n* = 22) and ALS patients (*n* = 29) at baseline, with no difference in the ALS cohort during the longitudinal follow up (four months) [7]. However, a recent pilot cross-sectional study performed by our group showed no difference between T1-w paraspinal muscles images of ALS patients (*n* = 10) and healthy controls (*n* = 9) [8].

The aim of the present study is to assess the role of paraspinal muscle MRI as a supportive diagnostic tool in ALS, especially during the early phase of the disease, by comparing the pattern and degree of muscle involvement between ALS patients, healthy controls, and other pathological conditions, such as inflammatory myopathies and lumbar radiculopathies.

## 2. Materials and Methods

### 2.1. Patients

In this cross-sectional study, we enrolled fourteen patients newly diagnosed with probable or definite ALS using the El Escorial criteria [6], between 1 January 2016 and 31 December 2018 at the IRCCS Mondino Foundation. Some subjects were recruited from a previous study [8]. Among the exclusion criteria were the inability to give informed consent, a contraindication to MRI, and respiratory failure impairing the ability to lie in supine position in the MRI scanner for the necessary time. We scored clinical severity by using the revised ALS Functional Rating Scale (ALS-FRSr) [9]. We also recruited the following subgroups of subjects for MRI analysis: age-matched healthy controls (HCs), patients with inflammatory myopathy, and patients affected by lumbar radiculopathy. The institute’s ethics committee (Ethics Committee Pavia) approved the study (number p-20170020469), and all subjects gave their written informed consent. The study was conducted in accordance with the Declaration of Helsinki.

### 2.2. MRI Data Analysis

Subjects underwent a 3T MRI (Siemens Skyra, Erlangen, Germany) lumbar spine examination that included an axial TSE T1 sequence (slice thickness = 10 mm; distance factor = 10%; TR/TE, 803/11 msec; NEX = 24; FoV = 230 mm; matrix = 512) and a coronal TSE T1 image (slice thickness, 5 mm; distance factor = 10%; TR/TE, 714/12 msec; NEX = 2; FoV = 400 mm; matrix = 512) centered on the paraspinal muscles, from the dorsal to the sacral region. The 16-channel spine coil was used. The MRIs were examined independently by two neuroradiologists with different experience in neuromuscular disease (A.P. with 20 years, and M.P. with two years of experience), blinded to clinical data. Each muscle (multifidus, longissimus dorsi, psoas) was visually graded for degree of fatty replacement using the Mercuri scale [10], i.e.:Stage 0: normal appearance;Stage 1: early moth-eaten appearance, with scattered small areas of increased signal intensity on the T1 MR sequence;Stage 2a: late moth-eaten appearance, with numerous discrete areas of increased signal intensity (MRI) with beginning confluence, comprising less than 30% of the volume of the individual muscle;Stage 2b: late moth-eaten appearance, with numerous discrete areas of increased signal intensity (MRI) with beginning confluence, comprising 30–60% of the volume of the individual muscle;Stage 3: washed-out appearance, fuzzy appearance due to confluent areas of increased signal intensity (MRI), with muscle still present at the periphery;Stage 4: end-stage appearance, muscle replaced by increased signal intensity (MRI) connective tissue and fat, with only a rim of distinguishable fascia and neurovascular structures.

### 2.3. Statistical Analysis

All data are reported as means, with ranges for quantitative variables, and percentages for categorical ones. In order to compare differences in MRI data among groups, chi square tests and non-parametric Kruskall–Wallis tests were performed. To test inter-rater reproducibility for the semiquantitative visual analysis of muscles between the two observers, Cohen’s Kappa was performed. *p*-values ≤ 0.05 were considered significant (two-sided). The statistical software STATA V.14 and R 3.6.0 were used for the analysis.

## 3. Results

Fourteen patients (males = 7 and females = 7; mean age = 61.3 years, range = 29–79) with definite (*n* = 2), or probable (*n* = 12) ALS, and eleven age-matched HCs (males = 8 and females = 3; mean age = 66.3 years, range = 48–76) were recruited. Ten patients (males = 5 and females = 5; mean age = 46 years, range = 26–62) with inflammatory myopathy, and nineteen (males = 13 and females = 6; mean age = 61.4 years, range = 44–75) with lumbar radiculopathy were also recruited as pathological controls. Clinical and demographic data are summarized in Table 1.

An adequate image quality was obtained in the paraspinal MRI studies of all patients and controls, and each muscle was graded using the Mercuri scale (Appendix A). Inter-rater reproducibility was assessed with coefficients with variation <5% for all regions of interest, specifically for multifidus (*k* = 0.935; *p* < 0.05), longissimus dorsi (*k* = 0.919; *p* < 0.05), iliocostal (*k* = 0.963; *p* < 0.05), and psoas (*k* = 0.957; *p* < 0.05).

No statistically significant differences in fatty replacement were shown for the multifidus (*p* = 0.1646), longissimus dorsi (*p* = 0.8958), or iliocostal muscles (*p* = 0.376) between the cohort subgroups, though ALS patients showed less fatty replacement of the psoas muscles, compared to patients with radiculopathy (*p* = 0.003). After stratifying ALS patients into spinal and bulbar onsets, we found statistically significant differences between the spinal ALS subgroup (*n* = 10) and bulbar ALS subgroup (*n* = 4) for the longissimus dorsi muscle (bulbar patients, *p* = 0.0245; radiculopathy *p* = 0.1335; myopathy *p* = 0.2575; HCs *p* = 0.1751), while no difference was found for multifidus (*p* = 0.1441), iliocostal (*p* = 0.0655), and psoas muscles (*p* = 0.0813) between the cohort subgroups. Figure 1 shows sample MRI scans.

No significant differences in muscle appearance were found between the right and left side in all patient subgroups.

## 4. Discussion and Conclusions

In the present study, we aimed to study the difference in MRI alterations of paraspinal muscles in T1-w sequences between ALS patients and groups of healthy controls, as well as groups with other pathological conditions. We focused our attention on paraspinal muscles because of their importance in staging ALS using El Escorial criteria. The evaluation of paraspinal muscles is usually performed with EMG (Electromyography) studies at thoracic level; however, paraspinal muscle MRI allows for a broader evaluation, including the analysis of single muscles at different levels, as well as the evaluation of deep muscles. The psoas muscle was also included in the evaluation, even though it is not classified as a paraspinal muscle, though having similar features and functions. The finding that ALS patients showed less fatty replacement of the psoas compared to patients with radiculopathy (*p* = 0.003) could be explained by its innervation by the lumbar plexus (L2–L4) motor fibres, which are often involved in degenerative radicular pathology. Interestingly, the significant difference in psoas fatty replacement between bulbar ALS patients and HCs (*p* < 0.05) that our group had previously found [8] has not been confirmed by this present study. Studies with larger cohorts of patients could help in further clarifying the involvement of the psoas. Conversely, no statistically significant differences were detected for the multifidus, longissimus dorsi, and iliocostal muscles between the four subgroups. The more pronounced fatty replacement of the longissimus dorsi in spinal onset ALS, compared to bulbar onset ALS (*p* = 0.0245), could indicate that the pathological process seems to be limited to a specific region in the former subgroup. No difference was observed when evaluating longissimus dorsi appearance between spinal ALS and other subgroups (radiculopathy *p* = 0.1335; myopathy *p* = 0.2575; HCs *p* = 0.1751). Other studies should be done in order to confirm that longissimus dorsi could be a potential candidate as a diagnostic ALS biomarker. Furthermore, after stratifying ALS patients by onset and comparing spinal onset ALS patients to other groups, no difference was found for other muscles (multifidus *p* = 0.1441, iliocostal *p* = 0.06553, and psoas *p* = 0.08132). Future studies on larger cohorts could help to clarify the role of paraspinal muscle MRI in the early diagnostic phase and differential diagnosis of ALS.

Literature on paraspinal MRI in ALS is scarce and heterogeneous in terms of study type (longitudinal [7], cross-sectional [8]), and MRI sequences (qualitative T1-w [8] and T2-w [7]). In the two cited studies, age-matched healthy controls were recruited [7,8] without recruiting patients with different pathologies. Jenkins et al. [7] proposed a fast protocol for T2-w whole-body muscle MRI, in order to longitudinally study the denervation in motor neuron disease. They found statistically significant differences in the T2-w images of paraspinal muscles between healthy controls (*n* = 22) and ALS patients (*n* = 29) at baseline, though no differences were observed during the longitudinal follow up (four months) in the ALS cohort [7]. T1-w sequences for chronic muscle alterations were not performed. In our explorative study [8], we found no differences between T1-w paraspinal muscles images of newly diagnosed ALS patients (*n* = 10) and healthy controls (*n* = 9), though we did not perform T2-w sequences.

Hence, we count the lack of T2-w sequences among the limitations of the present study, as they are crucial for evaluating early muscle changes (i.e., edema). Other limitations include the lack of quantitative muscular sequences; however, quantitative muscle analysis is beyond the scope of this pilot study. Moreover, a small cohort of patients was recruited, which reflects the low incidence of ALS. Lastly, patients affected by inflammatory myopathy were recruited in different phases of the disease, and the muscle damage between the patients could be significantly different in terms of fatty substitution.

In conclusion, paraspinal T1-w MRI could help in distinguishing spinal ALS patients from healthy controls and patients affected by other pathological conditions. Specifically, the study of the longissimus dorsi could play the role of a diagnostic ALS biomarker. To confirm these preliminary results, we plan to conduct further studies with a larger cohort of patients and in different homogeneous pathological conditions, with a more comprehensive protocol that includes both qualitative and quantitative T1-w and T2-w images, in order to evaluate both acute and chronic muscular alterations.

## Figures and Tables

**Figure 1 jcm-09-00934-f001:**
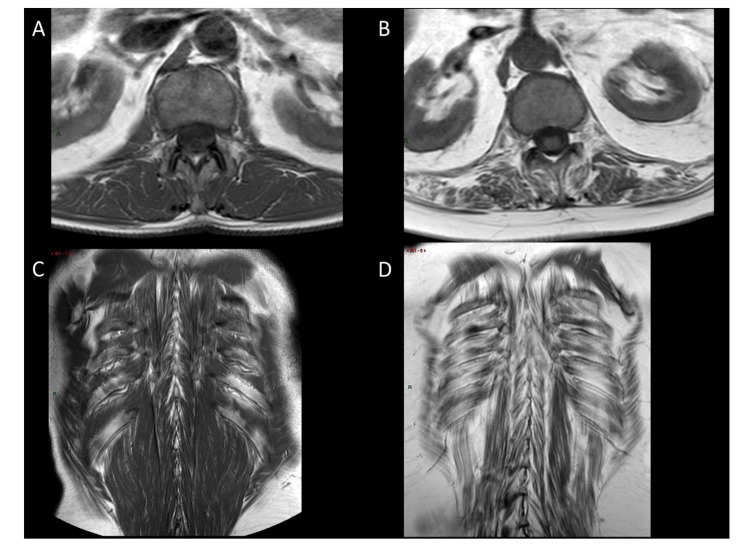
MRI scans of a healthy control (**A**,**C**—Healthy control (HC) #4) and an ALS patient (**B**,**D**—patient #10). Paraspinal axial (**A**—HC #4, **B**—patient #10), and coronal T1 (**C**—HC #4, **D**—patient #10) images show an extensive degree of fat replacement in the ALS patient compared to the HC.

**Table jcm-09-00934-t001a:** (**A**)

Data	
Gender, M/F	7/7
Age at onset, mean in years (range)	61.3 (29–79)
Time from onset to diagnosis, mean in months (range)	12 (3–26)
ALSFRSr at MRI, mean (range)	41.2 (33–47)

**Table ijerph-554968-t001b:** (**B**)

Patient ID	Gender	Age at Onset (Years)	Site of Onset	ALSFRSr at MRI
1	M	29	Spinal	47
2	F	52	Spinal	45
3	F	39	Spinal	38
4	M	74	Spinal	44
5	F	65	Bulbar	40
6	M	64	Spinal	33
7	F	55	Spinal	39
8	F	76	Bulbar	42
9	M	73	Bulbar	44
10	F	64	Spinal	41
11	M	56	Spinal	32
12	M	69	Spinal	46
13	F	63	Bulbar	44
14	M	79	Spinal	42

Abbreviations: M = male; F = female; ALS-FRSr = Amyotrophic Lateral Sclerosis Functional Rating Scale revised.

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
