# Peer review of "MRI Study of Paraspinal Muscles in Patients with Amyotrophic Lateral Sclerosis (ALS)"

_jcm, 2020, doi:10.3390/jcm9040934_

Round 1
Reviewer 1 Report
This is a very interesting study, by the originality and in an attempt to better understand a rare and heterogeneous disease.
I'd like to be explained the reason for the differences between the current study and the previous.
I hope that after the correction of the limitations of this study, better dirty knowledge in ALS
Author Response
We thank the reviewer for the positive and stimulating comments.
Other points:
- I'd like to be explained the reason for the differences between the current study and the previous. The significant difference in psoas fatty replacement between bulbar ALS patients and HCs (p<0,05) that our group had previously found has not been confirmed by this present study. The discrepancy is likely due to the small number of patients involved in both the studies. Further
studies with larger cohorts of patients could help in clarifying the involvement of the psoas, that is a spinal muscle, and it could be studied to support the diagnosis in bulbar patients.
Reviewer 2 Report
The authors presented a single center study to evaluate paraspinal muscle MRI as a diagnostic biomarker in amyotrophic lateral sclerosis (ALS). In conclusion, the authors considered the MRI of M. longissimus dorsi to play a role as a diagnostic biomarker for ALS.
In my opinion, the conclusion is premature. The approach of establishing MRI to confirm diagnosis of ALS is an interesting idea to minimize invasiveness of test methods. First of all, the number of patients included is very low, and no clinical information of the control groups is given. Although control patients with lumbar radiculopathy were included, more patients with degenerative spinal muscles atrophy i.e. patients with spinal canal stenosis should be added as a control group. The method used (Mecuri Scale) is a rather subjective method for the measurement of muscle degeneration. A more objective and standardized way of assessment is necessary and would be i.e. a volume-based measurement. Thus, group size, method, and interpretation need to be optimized and study design reconsidered.
Author Response
We thank the reviewer for the insightful comments. We completely agree with the major criticism, in fact we have declared limitations of the study (group size, method, and controls) in the final part of the paper. For sure, the conclusion is premature given that this is a pilot study, and it will be necessary to confirm these preliminary results, conducting further studies in a larger cohort of patients and in different homogeneous pathological conditions, also with a comprehensive protocol with both qualitative and quantitative T1-w and T2-w images.
Reviewer 3 Report
This is a very interesting subject and of great clinical relevance. I do, however, have a few suggestions to make this paper more effective for sharing your findings:
1) Tabular representation of the results would be most helpful
2) I would like to know more about the MRI findings in non-ALS cases, both in the table suggested above and in the text
3) The text seems to contradict itself regarding the statistical significance in the longissimus between spinal ALS and other disease conditions. In the results section, you state it is statistically significant but in the discussion, you state it isn’t. Which is correct?
4) The MRI images are excellent and add a lot to presenting the content
5) The wording of the article is awkward and would benefit from editing for English language
Overall, this is clinically relevant work and would add greatly to the ongoing efforts to determine feasible biomarkers for ALS.
Author Response
We thank the reviewer for the good comments and the hints.
Other points:
1) Tabular representation of the results would be most helpful AND 2) I would like to know more about the MRI findings in non-ALS cases, both in the table suggested above and in the text. We have proposed all the results (Mercuri scale for each muscle) in the supplementary data.
4) The MRI images are excellent and add a lot to presenting the content. We have added MRI images of patient with myopathy and patient with radiculopathy in the supplementary data.
3) The text seems to contradict itself regarding the statistical significance in the longissimus between spinal ALS and other disease conditions. In the results section, you state it is statistically significant but in the discussion, you state it isn’t. Which is correct? We have modified the results section (pag. 10, line 138-143).
5) The wording of the article is awkward and would benefit from editing for English language. Done.